**Data Availability Statement:** Given the nature of the data (qualitative focus groups with individuals discussing their experiences of conducting specific

# Researchers' experiences of the design and conduct challenges associated with parallel-group cluster-randomised trials and views on a novel open-cohort design

Claire Surr[1]©*, Laura Marsden[2]©, Alys Griffiths[3], Sharon Cox[4], Jane Fossey[5], Adam Martin[6], A. Toby Prevost[7], Catherine Walshe[8], Rebecca Walwyn[2]©

1 Centre for Dementia Research, Leeds Beckett University, Leeds, United Kingdom, 2 Clinical Trials Research Unit, University of Leeds, Leeds, United Kingdom, 3 Institute of Population Health, University of Liverpool, Liverpool, United Kingdom, 4 Department of Behavioural Science and Health, UCL, London, United Kingdom, 5 Faculty of Health and Life Sciences, University of Exeter, Exeter, United Kingdom, 6 Academic Unit of Health Economics, University of Leeds, Leeds, United Kingdom, 7 Nightingale-Saunders Clinical Trials & Epidemiology Unit, Kings College London, London, United Kingdom, 8 International Observatory on End of Life Care, Lancaster University, Lancaster, United Kingdom

© These authors contributed equally to this work.

* c.a.surr@leedsbeckett.ac.uk

## Abstract

### Background

Two accepted designs exist for parallel-group cluster-randomised trials (CRTs). Closed-cohort designs follow the same individuals over time with a single recruitment period before randomisation, but face challenges in settings with high attrition. (Repeated) cross-sectional designs recruit at one or more timepoints before *and/or* after randomisation, collecting data from different individuals present in the cluster at these timepoints, but are unsuitable for assessment of individual change over time. An 'open-cohort' design allows individual follow-up with recruitment before and after cluster-randomisation, but little literature exists on acceptability to inform their use in CRTs.

### Aim

To document the views and experiences of expert trialists to identify:

a) Design and conduct challenges with established parallel-group CRT designs,

b) Perceptions of potential benefits and barriers to implementation of open-cohort CRTs,

c) Methods for minimising, and investigating the impact of, bias in open-cohort CRTs.

### Methods

Qualitative consultation via two expert workshops including triallists (n = 24) who had worked on CRTs over a range of settings. Workshop transcripts were analysed using Descriptive Thematic Analysis utilising inductive and deductive coding.

C-RCTs that are published), it is not possible to fully anonymise the data. Therefore, participants would be potentially identifiable if data were placed in a public repository or the data would need redacting to the point it would be unsuitable for further analysis. Our original participant information and consent documentation did not include information about or consent for sharing of the data to a publicly available repository. Therefore we have no legal basis to share in this way, given it would not be possible to fully anonymise the data. Any requests to access the data can be made via the University Research Ethics Sub-committee (URESC). Requests may be made via the Chair Prof. Ruth Robbins (R.Robbins@leedsbeckett.ac. uk) and the committee will undertake independent review of such requests.

**Funding:** RW (Grant holder) and CS received award funding in association with conducting this study from The Medical Research Council Methodology Research Programme grant number MR/P026761/1 https://www.ukri.org/councils/mrc/ The sponsors and funder played no role in study design, data collection and analysis, decision to publish or preparation of the manuscript.

**Competing interests:** The authors have declared that no competing interests exist.

**Abbreviations:** CC, Closed cohort; CRT, Cluster randomised trial; DCM, Dementia Care Mapping; ITT, Intention to treat; OC, Open cohort; R-CS, Repeated cross-sectional; RCT, Randomised controlled trial.

## Results

Two central organising concepts were developed. *Design and conduct challenges with established CRT designs* confirmed that current CRT designs are unable to deal with many of the complex research and intervention circumstances found in some trial settings (e.g. care homes). *Perceptions of potential benefits and barriers of open cohort designs* included themes on: approaches to recruitment; data collection; analysis; minimising/investigating the impact of bias; and how open-cohort designs might address or present CRT design challenges. Open-cohort designs were felt to provide a solution for some of the challenges current CRT designs present in some settings.

## Conclusions

Open-cohort CRT designs hold promise for addressing the challenges associated with standard CRT designs. Research is needed to provide clarity around definition and guidance on application.

## Introduction

Cluster-randomised trials (CRTs) randomise groups of individuals ("clusters") to different interventions or sequences of interventions within a trial, as opposed to individuals. They have become increasingly more common since their initial use in the 1980s [1], with their use increasing since the early 2000s when the first CONSORT extension to CRTs was published [2]. CRTs are widely used in settings where interventions are delivered in an attempt to change the culture, environment or general practices and to reduce contamination between arms. This frequently occurs in schools, care homes and healthcare settings including both primary and secondary care [3, 4]. CRTs are also a common choice for trials conducted in communities or villages in low- and middle-income countries [5], where cross-contamination between arms or logistical and administrative reasons mean a standard RCT design would be problematic [6].

Two widely accepted designs currently exist for parallel-group CRTs. Closed cohort (CC) designs follow the same individuals over time, with recruitment occurring just once prior to cluster-randomisation. (Repeated) cross-sectional (R-CS) designs allow for recruitment before and/or after cluster-randomisation at one or more discrete time points, collecting data from different individuals present in the cluster at these timepoints. Some individuals are potentially measured more than once [7] but repeated measurements on individuals are often not linked over time. Other designs exist, but they are currently not labelled and each requires their own methodological literature. The focus of this paper is on open cohort parallel-group CRTs.

CRTs with CC designs face challenges in settings with high attrition rates. Such settings include care home and palliative care settings, largely due to participant death or moving to another setting, as well as other settings such as prisons, where the presence of prisoners with shorter sentences similarly leads to high participant turnover. There are examples of CC CRTs in these settings where fewer than 50% of baseline participants were remaining at trial end, decreasing statistical power and potentially leading to attrition bias, consequently compromising internal and/or external validity [8–12].

To overcome expected high attrition rates, CRTs may intentionally avoid evaluation of long-term outcomes from the outset [13], for example by including alternative primary and secondary outcomes and selecting follow-up periods that minimise attrition. Care home and

palliative care trials may also use minimum life expectancy as participant inclusion criteria [14–16]. In these settings, trialists have noted the difficulty of choosing a suitable follow-up period, identifying a trade-off between the trial being long enough to implement the intervention and assess its sustainability, but short enough to minimise losses [9, 17, 18]. Anticipated attrition may, therefore, force adaptation of the research question when using a CC design, narrowing the target population to which inferences can be made. This is a concern as the research question should drive the trial design rather than vice-versa. Similarly, the R-CS design, due to its cross-sectional nature, is able to provide cluster-level inference at specific time points. It is generally unsuitable where the research questions involve an assessment of individual change over time.

## Motivating example for this study

To provide a clear rationale for the need to consider novel "open-cohort" trial designs we will present details of a motivating study, which exemplified how neither of the established designs were entirely suited to achieving the trial's objectives. The issues presented in this motivating example are not exclusive to this trial, but common to other CRTs in settings where the intervention operates at a cluster level and requires a period of follow-up that is likely to mean high study attrition. However, such design challenges largely remain unacknowledged in trial reporting.

The DCM-EPIC trial [19], was a parallel-group CRT with economic evaluation where clusters (care homes) were randomised to a Dementia Care Mapping (DCM) intervention plus usual care, or usual care only. DCM involves observation of care practice using a standardised tool, analysis of data and feedback of findings to the staff team. These are then turned into action plans for care home and individual resident level practice change and comprise one practice development cycle. Thus, the intervention comprised both individual and cluster level components, which had an overall aim of improving care quality, with the ultimate aim of impacting resident outcomes. The primary continuous endpoint was resident-level agitation assessed 16-months after cluster-randomisation. The 16-month timepoint was adopted since the intervention needs time to embed into practice, and this endpoint permitted 3 'cycles' of DCM. Data was also collected at 6-months post-randomisation.

Originally, DCM-EPIC had a CC design as individual change over time was of interest (Resident A, Fig 1). However, trial monitoring indicated that up to 50% of residents could be lost to follow-up by trial end (Resident B, Fig 1), predominantly due to death, with a

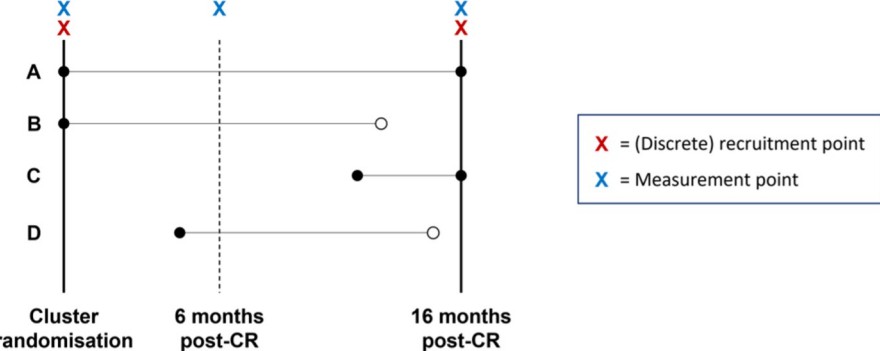

**Fig 1. Illustration of four different scenarios for residents in DCM-EPIC care homes.** Black circles denote a resident's presence; white circles denote the time a resident moved, withdrew from the trial, or died. Resident D was not recruited into the trial. CR = cluster randomisation.

smaller number of residents moving out of the care home. Continuation with the CC design would have led to lower statistical power and questionable external validity, as the sample of residents remaining at 16-months would not have been representative of the general care home population. A design change was approved by the funder and ethics panel which included recruitment of additional residents at 16-months from randomisation of the cluster (Resident C, Fig 1), and the primary endpoint instead utilised a cross-sectional analysis [19].

DCM-EPIC analysis and final trial reporting meant that although Resident D (Fig 1) was exposed to the cluster-level intervention, they were not recruited due to the timing and spacing of recruitment and measurement points. Thus, the number and timing of measurement and recruitment points is important in determining which residents are sampled. Given the original trial design an acceptable compromise for analysis and reporting was adopted (traditional R-CS analysis). However, neither the original CC design nor a R-CS approach made full use of the data collected from residents in DCM-EPIC, which had implications for statistical power, trial resource use, costs, and interpretation of results.

## A case for alternative trial designs

Whilst reviews have highlighted parallel-group CRTs using *both* closed cohort and cross-sectional approaches within the same trial [20, 21], this was to address different endpoints as opposed to a single design which unites the two approaches. CC and R-CS designs appear to be viewed as the only two possible, mutually exclusive options for parallel-group CRTs, forcing trialists to choose between them. To overcome the aforementioned issues in future, an ideal design would collect both CC (A) and cross-sectional (C) data, as well as data from CC participants lost to follow-up (B) and from those present in between baseline and final follow-up (D), all contributing to assessment of the same endpoint. This design, which allows for recruitment of individuals both before and following cluster-randomisation, and repeated measurements on individuals that crucially can be linked over time (unlike repeated cross-sectional samples), could be described as an "open cohort" or "dynamic cohort" design. We will refer to it as open-cohort (OC). The OC design leads to missing baseline data by design for participants recruited after randomisation.

However, there is little methodological literature published to inform OC designs for CRTs [22] or experience to suggest their acceptability as a valid trial design by trialists. Therefore, to further the utility of OC designs, this study is part of a larger MRC-funded study on 'OPen-cohorts in Institutional Settings: designs for Cluster-Randomised Trials' (OPIS-CRTs), which includes a literature review, user engagement, statistical development and evaluation, and practical guidance which aims to address these gaps. This paper reports on the user engagement component.

## Objectives

To document the views and experiences of expert trialists involved in parallel-group CRTs to identify:

1. Design and conduct challenges with established parallel-group CRT designs,

2. Perceptions of potential benefits and barriers to implementation of OC parallel-group CRTs,

3. Methods for minimising, and investigating the impact of, bias in OC parallel-group CRTs.

## Design and methods

This study adopted a qualitative expert consultation approach, through conduct of expert workshops.

### Expert workshops

Expert workshops are facilitated small group events that allow individuals with experience in the domain of focus, to actively participate in discussion and activities to achieve a particular outcome [23]. They provide an opportunity to gain immediate reaction to, and feedback on, presented information through a semi-structured approach to facilitating discussion, while permitting flexibility to respond to and explore issues that emerge during discussions [24].

Two workshops were held in 2019, the first with expert trialists who had worked on the DCM-EPIC CRT [25] (many of whom had also worked on other CRTs) and those who had recently conducted CRTs in care homes or hospices. The second included a broader group of expert trialists with experience of CRTs, with diverse professional backgrounds, working across a range of fields and service settings, none of whom had worked on DCM-EPIC.

### Recruitment and consent

Purposive and snowball sampling were used to identify workshop participants with relevant expertise, representative across a range of trial roles (e.g. chief investigator, statistician, health economist, academic or clinical researcher). Potential participants were identified through the research team's existing networks, approaching corresponding authors on relevant published studies identified in the literature review component of the larger study and by contacting Chief Investigators on current or recently completed, relevant trials listed on trial registers and databases of National Institute for Health and Care Research funded studies. Recruited participants were asked to suggest other individuals to approach, whose expertise would address any sampling gaps.

Inclusion criteria were:

1. Has taken part in a CRT (for Workshop 1 only, conducted in care homes or hospices).

2. Has an in-depth understanding of trial design and methods.

Participants were approached by e-mail by a member of the research team. They provided written informed consent to participate and were reimbursed for their travel but were not paid for attending.

### Data collection

The expert workshops took place on 1st May and 16th October 2019 respectively and were held face-to-face with the option to join by video conference where necessary. They were audio recorded and transcribed with 2-months of the workshop. Both workshops took place over a full day.

Each workshop was facilitated by five members of the study team [redacted] and consisted of short presentations on topics associated with the design and conduct of CRTs, including those with OC designs, followed by guided discussion. Topics included i) recruitment, bias and data collection, ii) the impact of intervention type and iii) intervention dose and exposure time. Workshop 1 focussed on CRT design and conduct where clusters are care homes or hospices; both workshops explored the use of OC designs as a potential alternative to established CC or R-CS designs. A list of potential challenges and solutions was generated. Those

identified in Workshop 1 were synthesised by the research team and taken forward to discussions in Workshop 2.

## Ethical issues

Leeds Beckett University ethical approval was obtained for the study. All participants provided written informed consent to participate. While confidentiality of individuals was maintained in analysis and presentation of the data, all workshop participants were given the opportunity to co-author this paper (subject to meeting co-authorship requirements) or to be named in the acknowledgements.

## Data analysis

Data were analysed during April and May 2021, using Descriptive Thematic Analysis [26] using both inductive and deductive coding. An initial set of deductive codes were developed based on the study objectives above (challenges, barriers and facilitators of different CRT designs). Inductive codes were developed associated with these deductive codes and where other topics of importance were identified in the data. Once coding was complete, codes were refined to form the themes and sub-themes presented. Coding and theme development was conducted by the first author and all transcripts and all coded data was reviewed by the second author to check meaning and corroborate themes. Disagreements were identified and discussed to reach agreement and refine themes accordingly.

# Findings

## Participants

Nine expert trialists participated in Workshop 1 (W1) and 15 in Workshop 2 (W2). Their demographics are presented in Table 1.

Two central organising concepts, four themes and four sub-themes (associated with two themes) were identified in the data (see Fig 2).

## Design and conduct challenges with established parallel-group CRT designs

Participants of both workshops identified a need to consider alternatives to CC and R-CS parallel-group CRT designs due to common challenges experienced. High loss to follow up was identified as being the greatest challenge faced. In care home and palliative care settings, high loss to follow up due to death or transfer out of the setting was expected and unavoidable:

> P8: So we're not looking at these [high loss to follow up rates] being unusual and. . .as the criteria get higher. . .to be admitted into residential care or nursing care. . .you're going to see people with much higher levels of frailty and other co-morbidities that. . .mean people are going to have less average time in a care home.
>
> W1
>
> P7: So of all of the trials [in a systematic review] that reported readings for their losses, 60% of them were due to death. So it's unavoidable. . .
>
> W1
>
> P6: . . .if you want to understand how an intervention works in practice I can't really see a reason–I'm exaggerating slightly–to have a closed cohort, because that's not what nursing

**Table 1. Demographics of expert workshop participants.**

|  | Workshop 1 | Workshop 2 | Total |
|---|---|---|---|
|  | n = 9 | n = 15 | N = 24 |
| **Sex, n** |  |  |  |
| Female | 6 | 10 | 16 |
| **Professional role(s)\*, n** |  |  |  |
| Statistician | 2 | 10 | 12 |
| Health Economist | 2 | 1 | 3 |
| Clinician | 2 | 1 | 3 |
| Academic subject expert | 2 | 1 | 3 |
| Trialist/ Methodologist | 1 | 2 | 3 |
| Funding panel member | - | 2 | 2 |
| **Previously worked on DCM-EPIC CRT?** |  |  |  |
| Yes | 5 | 0 | 5 |

\*Participants may fulfil more than one professional role

homes are. So that's not real life, that's not pragmatic. . .we know in any 12-month period, 30–40% of residents will change and often that is because of death. . .so I would really struggle with a. . .care home trial that explicitly excluded people who they expected to die. . .

W1

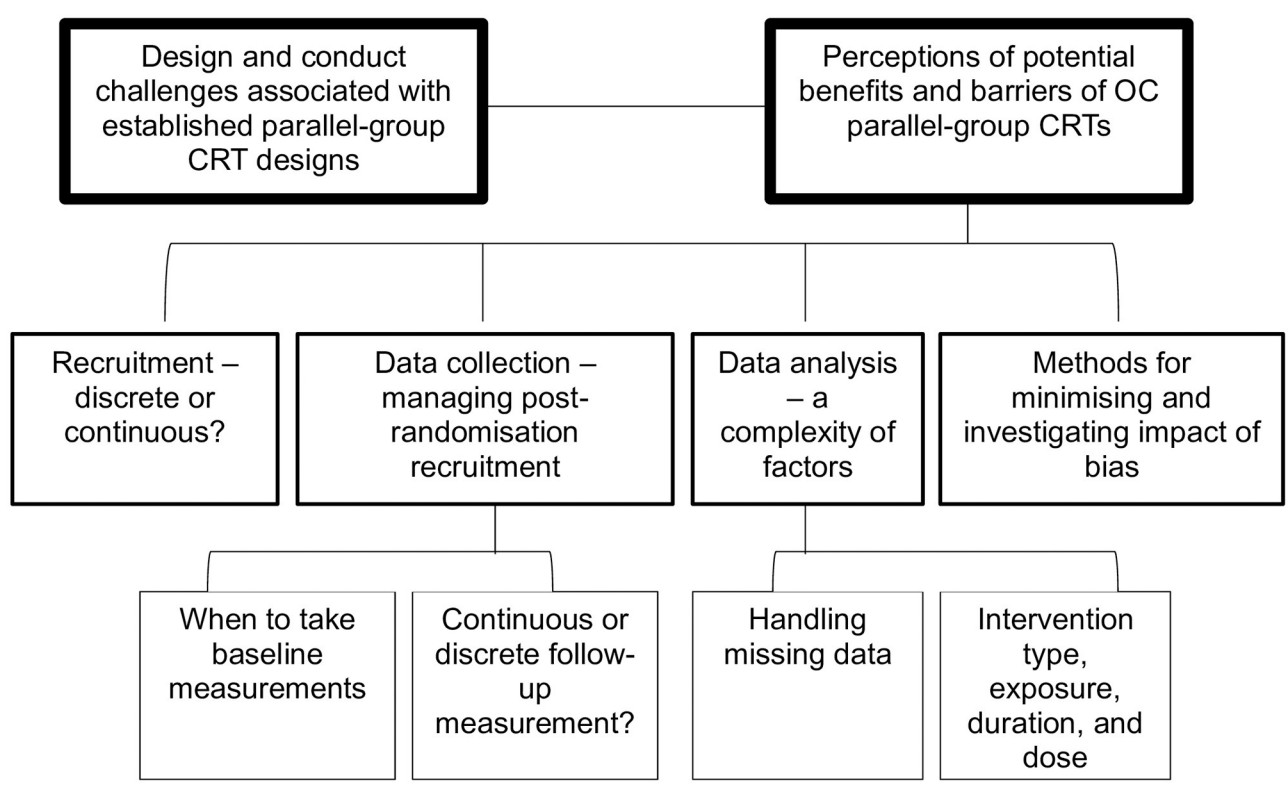

**Fig 2. Organising concepts, themes and sub-themes.**

Use of a CC design, where the cluster is inherently not closed, and in the face of high loss to follow-up, could result in missing data, loss of study power, problematic variation in cluster sizes and loss of entire clusters, introducing bias and raising issues for generalisability. If a R-CS design was adopted to address this, recruitment bias could potentially be introduced. Some participants may then be present in the dataset at more than one timepoint, without this being considered or accounted for within the analysis.

P5: In [trial name], I think the minimum was six [participants per cluster], to make the cluster viable. . . and other in homes sometimes, 40 people [in the cluster] how are you generalising data across them

W2

P1: at the moment we're excluding the people who die pretty quickly and. . .so we're only able to generalise [study findings] to the people who don't die particularly quickly.

P10: I suppose it depends whether your intervention is trying to target both groups and you might have different interventions that would work improving the quality of life for people who are expected to die quite quickly. . .But would those interventions also benefit the ones who are staying in the care home longer?

W1

A number of workshop participants reported using recruitment post-randomisation to address high loss to follow up in their CRTs, with some not modifying their analysis approach, or needing to conduct more than one analysis.

P9: There's the [Name] trial that we worked on. . .and it was the same thing, recruiting additional participants prior to the primary outcome at 12-months due to the drop out levels. . .I don't think it changed the analysis.

W1

P13: . . .a trial in nursing homes. . .we did have this problem about continuing to recruit participants if they were eligible during the trial because we wanted to increase our power and we ended up doing two types of analysis. One we called the cohort analysis which was the people that started at the beginning and then one we called the cross-sectional study which was just people that were [recruited post-randomisation] at 12-months.

W1

Thus, participants identified a need for alternative trial designs and appropriate, associated analysis methods.

Managing high loss to follow-up by limiting the follow-up period was identified as potentially appropriate for some interventions targeted at individuals in certain settings (i.e. palliative care) but may not be appropriate for other interventions and other settings. Short follow up raised problems for interventions, potentially at the cluster level, that need time to embed, or for effects to be realised, and meant sustainability could not be monitored.

P6: . . .if it's a palliative or end of life care intervention in a care home. . .you would be expecting people to die, so. . .we have a short follow up to try and capture as many people as

possible,...and...there would be an assumption–dose is really an important issue....if it doesn't work rapidly it's not worth doing

W1

Modifying standard trial designs to accommodate likely high loss to follow up, by adding or imposing strict eligibility criteria for example, was seen as sub-optimal.

## Perceptions of potential benefits and barriers of OC parallel-group CRTs

Workshop participants perceived that an OC design might provide a more efficient trial design, although this might not necessarily be the case if more complex analyses were planned. Designing a CRT as open-cohort from the outset was felt important to supporting appropriate decision-making and consideration of the range of design, implementation and analysis issues OC designs still raised.

P15: But I think from a design point of view to say that you were going to do this from the start . . . there were good reasons for doing it–so I think to set it up from the start is then a lot clearer about what people's expectations are.

W1

Only interventions that were truly cluster targeted were felt to be appropriate for an OC design.

P6: I think it really depends on the type of intervention and whether [there is] exposure to everybody . . . within the care home or whether some are . . . only delivered to some people . . . but the reason for choosing a cluster design is because you expect some leaking of the intervention out to everybody else.

W1

A range of design, conduct and analysis issues important for OC designs were identified as sub-themes (see Fig 2).

**Recruitment—Discrete or continuous?.** To address high loss to follow up, there was a strong consensus that either recruitment at one or more set time-points post-randomisation (discrete), or ongoing screening and recruitment at point of entry into the cluster (continuous) were particularly desirable aspects of an OC CRT design, due to the subsequent ability to recruit a more representative sample and improve generalisability. However, workshop participants highlighted practical challenges that recruitment post-randomisation (discrete or continuous) might present, with burden identified as the biggest challenge. This was particularly identified in sites which are less research ready/active, such as care homes, which generally then require considerable researcher resource to support recruitment.

P4: I think it also depends on whose burden it is. . . if we could have researchers going in and doing the majority of the research activity that's researcher burden. . .if you could minimally involve staff then it's potentially feasible to recruit

W1

P3: The study we've just finished it takes something like two and a half hours to drive from the most northerly care homes to the most southerly care homes so doing it in a day–no.

W1

Turnover within the trial setting impacting availability of trained staff to support ongoing screening and recruitment was also a potential barrier to continuous recruitment. This might be a particular challenge in sites with high staff turnover and few staff who have the expertise to support research activities, such as care homes, hospices, or other community settings.

P1: How are you going to manage the continual recruitment if you've . . .to go back in hoping there's going to be enough staff trained to keep that recruitment going?

W2

Participants in Workshop 1 agreed that, despite the potential benefits, continuous recruitment was unlikely to be desirable or practical in care homes or other similar settings, where research trained staff with resources to support research (for example NHS Research Nurses) were not available. Thus, recruitment at discrete timepoints probably provided the most appropriate option.

P7: I think in summary we're kind of saying we'd kind of like to do continual recruitment and data collection but it's probably not very practical.

P8: And I think not continuous but maybe at frequent timepoints and that would have to be decided based on the intervention and how long you need to follow up to be

P3: studies within the NHS . . . there might be research nurses on site every day. . .to recruit participants to a study, [you] have to recognise that [a researcher is] only going to be there intermittently [in a care home] so. . .unless the recruitment is actually being done by the people who are employed within the care home, . . . it's never going to be continuous, there is always going to be some intermittent nature to it.

W1

**Data collection—Managing post-randomisation recruitment.** The challenges identified with recruitment also applied to data collection. Discrete or continuous recruitment post-randomisation raised practical challenges for when and how it was appropriate to collect baseline and follow-up data.

Discrete or continuous recruitment post-randomisation raised practical challenges for identifying an appropriate baseline timepoint for each trial participant. For example, if baseline was the day of entry to the cluster, data covering the period prior to this was then not available, which might be necessary for a health economic analysis.

P6: often the primary analysis for health economics is cost utility [i.e.Quality Adjusted Life Years measurement], and that. . .requires collection of the array of costs. . .over time rather than a snapshot

W1

However, setting the baseline after a period of being present in the cluster might potentially expose a participant to the intervention prior to their baseline data collection. This also applied to instances of changed eligibility, for example, where a resident was ineligible at cluster randomisation but later became eligible.

P6: if you only want to collect data from people with advanced dementia, some of them may be present in the home when you start the study but will actually only become eligible-. . .at some point during the study.

P8: Yeah, that's another point. You could be exposed to the intervention before you're actually eligible for the trial.

W1

Following identification of an appropriate baseline, similar burden challenges as those for continuous recruitment were noted for continuous measurement, with some solutions offered such as reducing the number of outcomes collected or using routinely available data.

P7: So you might perhaps reduce down the number of outcomes that you actually collect in order to make that more feasible

W1

P4: but then. . .the collection of continuous data I think becomes unmanageable-. . .you're. . .going to need some staff involvement for proxy data. So it just becomes too excessive. Discrete timepoints is a good thing but perhaps slightly more frequently. . .it's a fine balance between collecting the data and creating too much burden.

W1

P5: . . .if there was some routine data that. . .could be standardised, collected in all care homes it wouldn't create additional burden.

W1

Workshop participants identified that the feasibility of continuous measurement might also depend on the nature of the outcomes being measured. They agreed that continuous recruitment did not have to necessitate continuous measurement; and measurement could instead be undertaken at discrete intervals. Continuous measurement might offer benefits for particular analysis approaches, but would potentially limit which outcomes could be assessed, especially when this data needs to be collected directly from participants or proxies.

P6: why would you necessarily have to do measurement at fixed times? Because in individually randomised trials, it's always sort of [a] floating time zero that's relevant to that individual isn't it?

P1: Fixed for individuals, but then obviously, that's a massive resource. . . Because you'd have to be back in each individual care home

P6: So, it depends on the outcome, how they are collected.

W2

P5: I mean I'm struggling to work out how you would ever have continuous data collection other than retrospectively going back to look for events because you wouldn't be collecting quality of life or patient reported outcomes continuously. They're always going to be at. . .discrete intervals.

W1

**Data analysis—A complexity of factors.** Participants identified a complexity of factors that must be considered when analysing CRT data with post-randomisation recruitment, thus highlighting potential challenges for OC designs. These included the handling of missing data and the exposure, duration and dose of intervention.

Discussions included whether death of a participant reflected missing data or should be considered an outcome, with different interpretations of this between the statistical and health economic analyses. Whether mortality was an outcome the intervention was expected to impact was identified as an important consideration, as this would influence how data missing due to death is treated.

> P2: I feel that inevitably death is an outcome and I'm nervous of disregarding that outcome and I guess it's a research question. If death is something that you try to avoid . . .[as] part of the research question then clearly, whether somebody dies or not is an outcome.

> P12: In health economics I don't think we'd say that death was missing data. For QALYS it obviously is zero, that's not missing. . .

> W1

> P3: To me, if you're interested in people's duration of time in the care home [vs when not in the care home] then you're not going to be imputing any of their data.. . . including the people who have died

> W1

Participants felt that statistical analysis methods needed to consider a range of factors: the specific estimand (e.g. intention to treat (ITT), per protocol, or something else entirely); the proposed intervention effect(s) (e.g. whether death or moving out of the cluster were potential outcomes); whether follow-up beyond a person's stay in the cluster was desirable (e.g. if outcomes are relevant to follow up if the participant moves care home) if feasible; how missing data was handled; and when imputation was appropriate. Thus, such decisions would need to be made on a trial-by-trial basis.

> P7: I think it depends on what is the estimand you're trying to capture. So if we are interested in an intention to treat estimand I would say yes. You have to go and follow them. . .if they left the nursing home.. . .Now it's a different matter if they die because I only use data that still exists. . .as opposed to counter factual data.. . .So I never impute dead people but I impute people that have been lost to follow on. And if I'm. . .only interested in people that remain exposed to the randomised treatment and that's a different matter.

> W1

> P3: . . .in our particular case if the intervention. . .[meant they were] less agitated then. . .there would be less care need for that person. So it may be informative the fact that they are having to move into a new care home.. . .also . . .going to a new care home is a way of rescuing them from the current environment that they're in and putting them into one that's more appropriate for their needs.

> P7: I agree but so long as you did this then you're no longer doing an intention to treat [analysis] and you've started to do a different type of analysis.

P3: And therefore you have a question as to whether your primary analysis should be the ITT one or not.

W1

Considerations for missing data were identified, such as whether it would logistically be possible to follow up those who left a cluster, or whether more realistically this data would need to be imputed. Reasons for leaving the cluster were felt to be important for the imputation model, but there were disagreements about what these might be.

P7: As long as they're alive, yes I would try to impute them.

P5: But what we require then is reasons why they've left their care homes so we can use that in information in the imputation process.

P7: . . .I would . . . construct an imputation model that tries to reconstruct the conditional distribution of the outcome given all their characteristics . . .then as secondary analysis we could do something statistically where we think that perhaps those people that moved out, moved out for a reason and maybe they're different from the ones that are staying . . . assuming that they're the same. . . is for me like a first safe bet.

P5: Do we really believe that's a safe bet though because the people who are missing–and there's a sizeable proportion of them–are really the same as those that aren't missing?

P3: . . .potentially–If you were to say that they were more close to the type of care home that they went into then that potentially would be one way of dealing with it and I'd be much happier.

P7: Yeah, correct. . .Why I'm saying it's the safe bet is because the rest are just even stronger assumptions. I'm with you that probably they're different but we don't really know how different they are

W1

Discrete and continuous recruitment post-randomisation also raised challenges for intervention type, exposure, duration and dose. Intervention type was identified as influencing exposure and dose and thus the appropriateness of an OC design.

Understanding potential dose-response relationships was felt an important consideration, particularly when participants might receive variable exposure to the intervention dependent on point of entry to, or exit from, the study.

P4: I think for the sort of studies we do,. . .dose is really challenging and that's what we struggle with. . .It's like a pharmaceutical study where we say 'do you know what? I've no idea whether you need 100 milligrams or 1000 milligrams. They've got the precursor studies so they know what the safe dose is–we don't tend to do that.

P6: It's a sort of logic model idea isn't it? Saying how much we think. You'd have to make a rational case for how much the dose you think may be effective and may have a physiological or psychological, social effect.

W1

Workshop participants discussed how variable exposure might result from intervention sustainability or decay effects, the point of joining the cluster and the length of time in the

cluster. This could be further complicated by whether a dose-response relationship is anticipated, learning curve and implementation delay effects, or intervention decay. All of these were felt to require consideration as part of the statistical analysis.

> P2: And [it] depends on if there's a dose-response relationship. . .whether the intervention is expected to have the same effect over a 3-month period as it would over a 6-month period or whether a 6-month period would be doubly effective
>
> P1: But if the intervention effect isn't sustained, if there's a waning. . ..
>
> P3: I guess this is also an outcome issue but . . .if you're collecting baseline of someone who joins the home at 6-months, the intervention's already established. . .
>
> W1
>
> P2: . . . for very onerous interventions that require staff to do lots of things well after the first initial period. If it is the same staff maybe they stop following guidelines? So maybe those individuals that are recruited to the trial much later get less exposed . . ..
>
> W1

Variable exposure was acknowledged as potentially further complicated by a clustering effect of the intervention.

> P1: And that average dose [assumed in statistical analysis] might be different in different clusters. So it might be tied up in the clustering effect as well. So how do you disentangle that
>
> W1

Looking beyond the challenges, workshop participants felt that an OC design required analyses to consider differing lengths of stay and to potentially link this to intervention effect, which was often not considered in other CRT designs.

> P5:. . .is it interesting to look at dose and time. We don't tend to look at that much individually in randomised trials do we? We just always stick with the ITT analysis and if we do look at dose it's always gonna be a supplementary thing that's not that interesting.
>
> P1: You see I think the clinicians are interested in that and when you fail to detect an intervention effect on your ITT analysis they want to know more about why did it work for some people, did we give enough of the dose? . . .
>
> P3: Also I think it's that person's contribution to the treatment effect. I don't think it's fair that somebody's contributing the same amount to the treatment effect if they're in the care home for a month as if they're in the care home for twelve months.
>
> W1

Yet, one workshop participant stated that even if an OC analysis was done, a more traditional analysis (i.e. CC or R-CS) should also be reported. This indicates a reticence, even among experts most likely to use OC designs, to move away from the more traditional analyses.

> P13. . . .one thing I would like to see is the traditional closed cohort presented either alongside or in a supplementary file. . .so what is the intervention now that we have this open

cohort? It will have less exposure. . .I would worry that we're analysing some average exposure which is very difficult to generalise to the general population unless you're telling us exactly what you mean by exposure–maximum exposure. The closed cohort I understand. . .because you know, for the duration of the cohort they were exposed to whatever. . .so maybe as an insurance policy I would like to see the more traditional analysis as well

W1

Even so, workshop participants commented that OC designs could create options for statistical analyses that specifically handle variable intervention exposure and time in the cluster.

P5: Is there a different challenge for the economic analysis that you're not observing people over the same length of time?

P6: It depends, if you expect there's an impact on survival you think.

P2: And depends on if there's a dose-response relationship. . .

P3: I think that's an issue for the statistics as much as it is the health economics.

P3: . . .actually I agree that that's not ideal but if you were to do an open cohort analysis you would allow for the fact that the people who have been in the care home over a period of time. . .

W1

**Methods for minimising, and investigating the impact of, bias.**   Workshop participants recognised that, due to the inability to not inform staff and residents of their intervention allocation, recruitment following randomisation included a risk of recruitment bias. This has the theoretical potential to impact willingness to consent, and lead to changes in staff and resident demographics. One participant (W1 P4) described this as becoming a 'magnet home', where certain staff or residents might choose to work/live, or which might alter the type of residents a care home felt able to admit/provide care for.

P15: . . .I think there are some issues there then about people's willingness to participate depending on how they've been randomised in the first place

W1

P5: depending on how long the recruitment is,. . .if your intervention is working, . . . indicators for that care home go up, then people will want to come to it, and you'll then have different people

W2

P1: If the. . .[care home] staff are better able to deliver care to more complex residents. . .it can. . .end up with people [moving in] who are more complex to start with. . .So, you've actually nullified any impact of the intervention. . .

W2

The potential for recruitment bias and differences in average time in the cluster between arms was noted as a problem with R-CS designs as well.

P2: I don't see why they aren't saying exactly the same about cross-sectional studies. Why are we able to publish those when we can't even look into those at this particular level of detail,. . .with this [open cohort design], at least we have some way of knowing the pattern of people who were recruited before randomisation versus the people who were recruited afterwards, which to me, makes it less worrying

W2

Workshop participants identified potential solutions to address recruitment bias including having tight inclusion and exclusion criteria, recruiting everyone eligible wherever possible, monitoring expected flow of participants into the cluster, and using blinded recruiters. Alternatively, using anonymised routine data that do not require individual consent was identified as a solution.

P10: I think you'd have to have an absolutely rock solid, objective entry criteria for the study wouldn't you?. . .I suppose the other way would be . . .somehow just use your routine data so that you didn't have to get. . . individual consent

W1

P9: So, we've done a kind of, open cohort. . .in emergency settings. And, we've been able to have a good control in estimating the numbers of people coming through and hence knowing that we've always got the right proportion [consenting per arm], that the characteristics of the proportions remains similar over time. . .

P3: One of the ways of preventing it, is just to recruit everybody at the cluster. . .

W2

## Discussion

This study is the first to consider, with expert trialists, the challenges of conducting parallel-group CRTs in institutional settings and their perspectives on a novel OC CRT design as an alternative. Workshop participants identified challenges associated with conducting parallel-group CRTs using established designs, with the predominant problem being expected large loss to follow up in some settings such as care homes and hospices. This reflected our experiences in the motivating case for this study, the DCM-EPIC CRT. Participants could generally recognise the value of OC designs but posed several questions around if and when an OC design might be appropriate.

While one of the primary features of OC CRTs, recruitment post-randomisation, was felt to be a key strength of the design, practical, methodological and statistical issues related to the feasibility of continuous recruitment were identified. Discrete recruitment points post-randomisation were felt to offer a solution to practical challenges associated with the resource intensive nature of recruitment, although palliative care trials have found resources and workload associated with this present a challenge for recruitment over longer periods [27]. In this study participants felt issues of potential sample bias and differential exposure to the intervention were less easily solved. While studies have considered ways in which allocation techniques can help to address balance at baseline in CRTs [28], there is less evidence related to this for recruitment post-randomisation. Use of masked or independent recruiters, [29, 30] and objective eligibility criteria [31] have been suggested as methods to limit the risk of recruitment bias, and baseline testing [32] and reporting of appropriate information [33] as methods for measuring it, however, further research that can address this gap is required.

Associated with recruitment post-randomisation was the related challenges for data collection including identification of baseline, timing of follow-up and implications for resources. Participants identified that routine data might address some, but not all, of these challenges. Routine or minimum datasets are readily available in some countries and settings, although concerns have been raised about the quality, completeness [34], comparability [35] and scope [36, 37] of data available for addressing research questions. However, in others for example UK care homes, there is no standardised method for capture of such data [38] and existing datasets may be fragmented [39]. Data linkage between social care and health data sets can be challenging and require specialist skill and resource [40, 41]. Thus, use of routine data may be realistic currently only for some outcomes, but holds future promise.

There were significant differences of opinion of workshop participants around approaches to statistical and health economic analysis and handling of missing data that require further exploration, to provide clearer guidance to statisticians and health economists around estimands, analysis methods and the assumptions underpinning these. Previous studies have identified challenges in incorporating cross-sectional data into health economic analyses [42]. Finally, learning curves and decay effects of the intervention and considerations of intervention sustainability identified in all CRTs remain challenges an OC design would need to address; this is likely to depend on the type of intervention and the level at which it is delivered. CRT analyses are often too simplistic, with intervention effects assumed constant following their implementation [43, 44]. Whilst tools already exist to encourage trialists to report details of how intervention drift was mitigated (e.g. Template for intervention description and replication (TIDieR)) [45], there is still a lot more to learn regarding intervention dose [46], and more focused research in this area is required in general for CRTs evaluating complex interventions. Questions also remain around how to determine the sample size for an OC design. Whilst Kasza [22] recently provided the first framework for this, including design effects, sample size formulae for specific sampling schemes and an R Shiny app for users, only three sampling schemes were proposed and are therefore not likely to be sufficient for all types of open cohort design. Further work is therefore required in this area for these designs to be readily adopted by trialists.

## Limitations

This is the first study to explore this topic with expert trialists. One limitation is that all trialists were UK-based and so the study does not include an international perspective. While workshop participants did represent the full range of trialist roles, it was weighted towards statisticians and to those working predominantly in care home trials.

Future work should include a literature review to assess the use of open-cohort designs within CRTs to date including how trial design might be influenced by the intervention, outcome type and setting. Clearer definition of OC CRTs as a study design is required, including guidance on when such designs are appropriate. It may be that in some situations, for example, more frequent measurement and an R-CS design is sufficient. Statistical development and evaluation should also be carried out to provide clear guidance on analysis approaches in OC CRT studies. This may include when it might be appropriate to exclude individuals from the analysis due to their limited exposure to the intervention.

## Considerations for researchers, funders and journal editors

Considerations for researchers, research funders and journal editors and reviewers
Researchers should:

- Openly acknowledge and critically address the challenges with conducting parallel group CRTs in populations or settings with unavoidably high attrition in grant applications and reporting of CRTs

- Propose appropriate, potentially non-traditional trial designs and analysis methods for trials in such settings

- Conduct methodological research to inform the development of guidance on OC and other potential non-traditional CRT designs

   Research funders should:

- Actively encourage researchers to acknowledge the methodological challenges associated with conducting CRTs in settings with unavoidable high attrition

- Be open to considering grant applications that adopt alternative trial designs, such as OC, to meet these challenges, including reasonable requests for additional resources that such designs may require

- Fund methodological research into alternative trial designs

   Journal editors and reviewers should:

- Actively encourage the open reporting of methodological challenges to conducting CRTs in settings with unavoidably high attrition, and the successes and challenges associated with approaches adopted to address these

- Publish studies that adopt non-traditional trial designs where they meet required markers of quality

- Publish research that advances methodological knowledge on OC and other non-traditional trial designs.

## Conclusions

OC CRT designs hold promise for addressing some of the challenges associated with standard CRT designs. However, there currently remains limited research on such designs to provide clarity around definition and guidance on their application.

## Acknowledgments

We would like to thank the following people for their contributions to the expert workshops: Professor Amanda Farrin, Professor Steph Taylor, Dr Clemence Leyrat, Prof Sally Kerry, Dr David Meads, Professor Alan Montgomery, Prof Anne Forster, Liz Graham.

## Author Contributions

**Conceptualization:** Claire Surr, Laura Marsden, Rebecca Walwyn.

**Data curation:** Laura Marsden, Alys Griffiths, Sharon Cox, Jane Fossey, Adam Martin, A. Toby Prevost, Catherine Walshe, Rebecca Walwyn.

**Formal analysis:** Claire Surr, Laura Marsden.

**Funding acquisition:** Claire Surr, Rebecca Walwyn.

**Investigation:** Claire Surr, Laura Marsden, Rebecca Walwyn.

**Methodology:** Claire Surr, Rebecca Walwyn.

**Project administration:** Rebecca Walwyn.

**Supervision:** Rebecca Walwyn.

**Writing – original draft:** Claire Surr, Laura Marsden, Rebecca Walwyn.

**Writing – review & editing:** Alys Griffiths, Sharon Cox, Jane Fossey, Adam Martin, A. Toby Prevost, Catherine Walshe.

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
