## [Decision Letter · Decision Letter 0]

31 Oct 2023

PONE-D-23-17325Researchers’ experiences of the design and conduct challenges associated with parallel-group cluster-randomised trials and views on a novel open-cohort designPLOS ONE

Dear Dr. Surr,

Thank you for submitting your manuscript to PLOS ONE. After careful consideration, we feel that it has merit but does not fully meet PLOS ONE’s publication criteria as it currently stands. Therefore, we invite you to submit a revised version of the manuscript that addresses the points raised during the review process.

We look forward to receiving your revised manuscript.

Kind regards,

Krit Pongpirul, MD, MPH, PhD.

Academic Editor

PLOS ONE

Journal Requirements:

Reviewers' comments:

Reviewer's Responses to Questions

**Comments to the Author**

1. Is the manuscript technically sound, and do the data support the conclusions?

Reviewer #1: Yes

Reviewer #2: Yes

2. Has the statistical analysis been performed appropriately and rigorously? 

Reviewer #1: N/A

Reviewer #2: N/A

3. Have the authors made all data underlying the findings in their manuscript fully available?

Reviewer #1: Yes

Reviewer #2: No

4. Is the manuscript presented in an intelligible fashion and written in standard English?

Reviewer #1: Yes

Reviewer #2: Yes

5. Review Comments to the Author

Reviewer #1: This is a well written manuscript which details qualitative findings from a group of trial experts on discussions of design limitations and potential of open cohort designs. While the study may have benefited from wider perspectives which would have been possible if done later, it is an interesting paper that adds important points to the literature and the consideration of trial methods.

Of note the authors should proofread carefully as trial is spelled trail in their abstract

Reviewer #2: This is an interesting paper which invites thinking differently the way a cluster randomized trial in nursing homes could be planned and managed. I have been really interest in reading it. I would have the following comments:

- Page 4, second paragraph: authors claim that there are two options which are the closed cohort design and the repeated cross-sectional one. So they seem to ignore the design in which there exists a continuous recruitment with an individual follow-up. This may be the case when nursing homes are recruited and randomized and then residents are recruited once an event occurs, such as an infection. This was for instance the case in Boocvar et al. J Am Geriatr Soc 2020;68:2329-2335. Why did authors ignore this design? Actually, it can be viewed as an open cohort design provided the recruitment is continuous. But I suggest adding some discussion on it.

- Page5, line 87: a high attrition rate is also a source of bias. This should be acknowledged.

- Page 5, line 96: adapting the research question to overcome the attrition problem is a highly debatable approach. I suggest that authors assert that it is surely not the right way to manage attrition.

- Figure 1. Resident D is indeed not taken into account when analyzing data in a R-CS analysis. That is true. However, do we loss many data? How numerous are residents D? This is never discussed in the paper. My question is: is it better to invest a lot (time, resources, etc) to catch residents D? Or is it better to miss them, and conduct a R-CS trial? This is of major importance, in my opinion, and this should be, at least, discussed in the last section of the paper.

- Page 8: authors define what they call an “open cohort” or “dynamic cohort” design. Actually, a (repeated) cross-sectional design can also be viewed as a specific type of open cohort, in which participants B and D would not be included for the assessment, except if there was an assessment at month 6). This could be acknowledged.

- Page 13 Table 2. This table, on its own, is very difficult to be understood. Personally, it is only after having read the full text that I understood it. So I suggest that authors make it more clear by completing it, or that they drop it.

- Page 15 – P6: I fully agree with the participant who asserted that nursing homes are not closed cohort. And I think that this idea should be more underlined than it is presently.

- Page 15 line 300: again, loss to follow-up may induce bias, not just loss of power and generalizability.

- Page 16, line 336: I disagree with the idea that to manage loss to follow-up, trialists can limit the follow-up period. This is, obviously, mathematically true. However, this changes the research question, by changing the time frame of the outcome. It is not only a question of sustainability. It really depends on the date at which the primary outcome is assessed. This is of major importance, and this should be more acknowledged (cf previous comment on page 5), although authors say that participants considered it was “sub-optimal” (page 17).

- Page 23, line 511: I am wondering whether all economists (I am not an economist) agree that for QALYS, death means zero. Is there any reference for that?

- Page 29-30: you may decide, in the eligibility criteria, that a resident, to be included in the statistical analysis, should have spent at least 2 months, for instance, in the nursing home. Nothing is said about that. Do we have to take into account all the residents, systematically? Or are we allowed to specify some selection criteria, notably criteria related to the exposition (or lack of exposition) to the intervention, due to a small period of time in the nursing home?

- Page 31, line 686: as far as I know, people join a care home depending on whether there are some available places, and they generally join a care home which is not far from their personal home. So, I do not believe at all that some resident may chose their care home depending on the group they have been allocated to. For me it is a purely theoretical risk.

- Discussion: if one intends to conduct an open-cohort design, how sample size could be determined? I suspect that many parameters had to be a priori specified (such as the proportion of participants who disappear), and surely, a lot of work has to be done on that topic. This should be discussed.

Minor comments

- Page 2 – line 5: « trail » should be trial

- Page 4 – line 62: clusters can be allocated to groups or to sequences (cross-over or stepped-wedge cluster randomized trials)

6. PLOS authors have the option to publish the peer review history of their article (what does this mean?). If published, this will include your full peer review and any attached files.

Reviewer #1: No

Reviewer #2: No

---

## [Author Response · Author response to Decision Letter 0]

27 Nov 2023

November 2023

Dear Editor, 

We are delighted that you feel our paper has merit and have invited us to submit a revised version of the manuscript entitled “Researchers’ experiences of the design and conduct challenges associated with parallel-group cluster-randomised trials and views on a novel open-cohort design”. Enclosed is a revised manuscript, addressing the comments from the reviewers. Changes are tracked and below are point-by-point responses to each of the issues raised.

1. Reviewer #1: This is a well written manuscript which details qualitative findings from a group of trial experts on discussions of design limitations and potential of open cohort designs. While the study may have benefited from wider perspectives which would have been possible if done later, it is an interesting paper that adds important points to the literature and the consideration of trial methods. Of note the authors should proofread carefully as trial is spelled trail in their abstract

We thank the reviewer and agree that that is scope to do further engagement work here as the methodology progresses from our early-stage work. The typo in the abstract has been corrected and we have proof-read the remainder of the manuscript. 

2. Reviewer #2: This is an interesting paper which invites thinking differently the way a cluster randomized trial in nursing homes could be planned and managed. I have been really interest in reading it.

We thank the reviewer. 

3. Reviewer #2: Page 4, second paragraph: authors claim that there are two options which are the closed cohort design and the repeated cross-sectional one. So they seem to ignore the design in which there exists a continuous recruitment with an individual follow-up. This may be the case when nursing homes are recruited and randomized and then residents are recruited once an event occurs, such as an infection. This was for instance the case in Boocvar et al. J Am Geriatr Soc 2020;68:2329-2335. Why did authors ignore this design? Actually, it can be viewed as an open cohort design provided the recruitment is continuous. But I suggest adding some discussion on it.

We agree that this is an important and commonly used alternative parallel-group cluster-randomised design that deserves a methodological literature associated with it. We were starting with the two designs that appear in the statistical trial methodology literature and are therefore widely accepted options. We have a further paper in preparation that looks at the full range of options, including the one mentioned by the reviewer. We have added the following sentence to page 4, paragraph 2: “Other designs exist, but they are currently not labelled and each requires their own methodological literature. The focus of this paper is on open cohort parallel-group CRTs.” 

4. Reviewer #2: Page5, line 87: a high attrition rate is also a source of bias. This should be acknowledged.

We agree. We have revised the sentence as follows: “There are examples of CC CRTs in these settings where fewer than 50% of baseline participants were remaining at trial end, decreasing statistical power and potentially leading to attrition bias, consequently compromising internal and/or external validity.”

5. Reviewer #2: Page 5, line 96: adapting the research question to overcome the attrition problem is a highly debatable approach. I suggest that authors assert that it is surely not the right way to manage attrition.

We agree and did not make this clear enough so we have revised as follows: “Anticipated attrition may, therefore, force adaptation of the research question when using a CC design, narrowing the target population to which inferences can be made. This is a concern as the research question should drive the trial design rather than vice-versa.”

6. Reviewer #2: Figure 1. Resident D is indeed not taken into account when analyzing data in a R-CS analysis. That is true. However, do we loss many data? How numerous are residents D? This is never discussed in the paper. My question is: is it better to invest a lot (time, resources, etc) to catch residents D? Or is it better to miss them, and conduct a R-CS trial? This is of major importance, in my opinion, and this should be, at least, discussed in the last section of the paper.

The number of Resident D’s in a trial will vary depending on the turnover rate and the frequency of measurements. As such, it will be setting and trial specific. In the case of care homes, there is a high chance of turnover in the first months after an individual joins the cluster for particular types of individual. Conducting a R-CS trial and missing them has the potential to miss this group of individuals and compromise the external validity of the trial. The importance of missing this group of individuals will depend on the research question – if it is sufficient to assess change at a population-level then a R-CS design would be sufficient, but if assessment of individual change is desired, it may not be. This last point is made in the last paragraph on page 5 as “Similarly, the R-CS design, due to its cross-sectional nature, is able to provide cluster-level inference at specific time points. It is generally unsuitable where the research question involves an assessment of individual change over time.” An additional point is now made in the first paragraph on page 36 as “Clearer definition of OC CRTs as a study design is required, including guidance on when such designs are appropriate. It may be that in some situations, for example, more frequent measurement and an R-CS design is sufficient.”

7. Reviewer #2: Page 8: authors define what they call an “open cohort” or “dynamic cohort” design. Actually, a (repeated) cross-sectional design can also be viewed as a specific type of open cohort, in which participants B and D would not be included for the assessment, except if there was an assessment at month 6). This could be acknowledged.

We agree that there are many common features between a R-CS design and an OC design. This paper highlights the need for very precise definitions. We believe that the main differentiator is that in R-CS designs, repeated measurements are purely cross-sectional; it is not possible to link repeated measurements from the same individual over time. We have added some clarification around this in the first paragraph on page 8 as “This design, which allows for recruitment of individuals both before and following cluster-randomisation, and repeated measurements on individuals that crucially can be linked over time (unlike repeated cross-sectional samples), could be described as an “open cohort” or “dynamic cohort” design.”

8. Reviewer #2: Page 13 Table 2. This table, on its own, is very difficult to be understood. Personally, it is only after having read the full text that I understood it. So I suggest that authors make it more clear by completing it, or that they drop it.

We have presented this information as a figure now – see Figure 2, which we summarises the organising concepts, their themes and sub-themes and their relationship more clearly. 

9. Reviewer #2: Page 15 – P6: I fully agree with the participant who asserted that nursing homes are not closed cohort. And I think that this idea should be more underlined than it is presently.

We agree. We have added greater emphasis to the point made by the participant in the third paragraph on page 15 as follows “Use of a CC design, where the cluster is inherently not closed, and in the face of high loss to follow-up, could result in missing data, loss of study power, problematic variation in cluster sizes and loss of entire clusters, introducing bias and raising issues for generalisability.”

10. Reviewer #2: Page 15 line 300: again, loss to follow-up may induce bias, not just loss of power and generalizability.

We have added this to the third paragraph on page 15 as “introducing bias and raising issues for generalisability.”

11. Reviewer #2: Page 16, line 336: I disagree with the idea that to manage loss to follow-up, trialists can limit the follow-up period. This is, obviously, mathematically true. However, this changes the research question, by changing the time frame of the outcome. It is not only a question of sustainability. It really depends on the date at which the primary outcome is assessed. This is of major importance, and this should be more acknowledged (cf previous comment on page 5), although authors say that participants considered it was “sub-optimal” (page 17).

We agree that in many cases limiting the follow-up period is undesirable. We have tried to clarify the issue as follows “Managing high loss to follow-up by limiting the follow-up period was identified as potentially appropriate for some interventions targeted at individuals in certain settings (i.e. palliative care) but may not be appropriate for other interventions and other settings. Short follow up raised problems for interventions, potentially at the cluster level, that need time to embed, or for effects to be realised, and meant sustainability could not be monitored.”

12. Reviewer #2: Page 23, line 511: I am wondering whether all economists (I am not an economist) agree that for QALYS, death means zero. Is there any reference for that?

Mathematically when calculating QALYs death must equate to zero since QALYS=HRQoL*LYs. So whatever value given to HRQoL, where LYs (life years) = 0 when dead, QALYS must also = 0. We have not included a reference to support this, given this point is mentioned in the findings where adding citations is not appropriate. 

13. Reviewer #2: Page 29-30: you may decide, in the eligibility criteria, that a resident, to be included in the statistical analysis, should have spent at least 2 months, for instance, in the nursing home. Nothing is said about that. Do we have to take into account all the residents, systematically? Or are we allowed to specify some selection criteria, notably criteria related to the exposition (or lack of exposition) to the intervention, due to a small period of time in the nursing home?

This was not raised by participants but we agree that it is an interesting point. So we have added the following to the discussion in the second paragraph on page 36 “Statistical development and evaluation should also be carried out to provide clear guidance on analysis approaches in OC CRT studies. This may include when it might be appropriate to exclude individuals from the analysis due to their limited exposure to the intervention.” 

14. Reviewer #2: Page 31, line 686: as far as I know, people join a care home depending on whether there are some available places, and they generally join a care home which is not far from their personal home. So, I do not believe at all that some resident may choose their care home depending on the group they have been allocated to. For me it is a purely theoretical risk.

We have made it clearer that this is a theoretical risk in the third paragraph on page 31 as “This has the theoretical potential to impact willingness to consent, and lead to changes in staff and resident demographics.”

15. Reviewer #2: Discussion: if one intends to conduct an open-cohort design, how sample size could be determined? I suspect that many parameters had to be a priori specified (such as the proportion of participants who disappear), and surely, a lot of work has to be done on that topic. This should be discussed.

We agree this is important to highlight, so have added the following to the discussion on page 35 “Questions also remain around how to determine the sample size for an OC design. Whilst Kasza (22) recently provided the first framework for this, including design effects, sample size formulae for specific sampling schemes and an R Shiny app for users, only three sampling schemes were proposed and are therefore not likely to be sufficient for all types of open cohort design. Further work is therefore required in this area for these designs to be readily adopted by trialists.”

16. Reviewer #2: Page 2 – line 5: « trail » should be trial.

Corrected. 

17. Reviewer #2: Page 4 – line 62: clusters can be allocated to groups or to sequences (cross-over or stepped-wedge cluster randomized trials)

Amended to “Cluster-randomised trials (CRTs) randomise groups of individuals (“clusters”) to different interventions or sequences of interventions within a trial, as opposed to individuals.”

Yours sincerely,

Professor Claire Surr on behalf of the co-authors

---

## [Decision Letter · Decision Letter 1]

2 Jan 2024

Researchers’ experiences of the design and conduct challenges associated with parallel-group cluster-randomised trials and views on a novel open-cohort design

PONE-D-23-17325R1

Dear Dr. Surr,

We’re pleased to inform you that your manuscript has been judged scientifically suitable for publication and will be formally accepted for publication once it meets all outstanding technical requirements.

Kind regards,

Krit Pongpirul, MD, MPH, PhD.

Academic Editor

PLOS ONE

Additional Editor Comments (optional):

Your responses to the comments from both reviewers are satisfactory. Happy New Year.

Reviewers' comments:

Reviewer's Responses to Questions

**Comments to the Author**

1. If the authors have adequately addressed your comments raised in a previous round of review and you feel that this manuscript is now acceptable for publication, you may indicate that here to bypass the “Comments to the Author” section, enter your conflict of interest statement in the “Confidential to Editor” section, and submit your "Accept" recommendation.

Reviewer #2: All comments have been addressed

2. Is the manuscript technically sound, and do the data support the conclusions?

Reviewer #2: Yes

3. Has the statistical analysis been performed appropriately and rigorously? 

Reviewer #2: N/A

4. Have the authors made all data underlying the findings in their manuscript fully available?

Reviewer #2: Yes

5. Is the manuscript presented in an intelligible fashion and written in standard English?

Reviewer #2: Yes

6. Review Comments to the Author

Reviewer #2: I thank the authors for their answers. All my comments have been answered in a satisfactorily and convincing way.

7. PLOS authors have the option to publish the peer review history of their article (what does this mean?). If published, this will include your full peer review and any attached files.

Reviewer #2: **Yes: **Prof Bruno Giraudeau

---

## [Editor Report · Acceptance letter]

15 Feb 2024

PONE-D-23-17325R1 

PLOS ONE

Dear Dr. Surr, 

I'm pleased to inform you that your manuscript has been deemed suitable for publication in PLOS ONE. Congratulations! Your manuscript is now being handed over to our production team.

Kind regards, 

on behalf of

Assoc. Prof. Dr. Krit Pongpirul 

Academic Editor

PLOS ONE